# Managerial Strategies for Long-Term Care Organization Professionals: COVID-19 Pandemic Impacts

**Ana Dias** [1],*[ID]**, Annibal Scavarda** [2][ID]**, Augusto Reis** [1][ID]**, Haydee Silveira** [3][ID] **and Nelson Francisco Favilla Ebecken** [3]

1   Production Engineering Department, Federal Center for Technological Education Celso Suckow da Fonseca-CEFET/RJ, Rio de Janeiro 20271-110, Brazil; professor.augusto.reis@gmail.com
2   Production Engineering Department, Federal University of the State of Rio de Janeiro-UNIRIO, Rio de Janeiro 22290-240, Brazil; annibal.scavarda@unirio.br
3   Civil Engineering Department, Federal University of Rio de Janeiro-COPPE-UFRJ, Rio de Janeiro 21941-450, Brazil; haydee.batista@coc.ufrj.br (H.S.); nelson@ntt.ufrj.br (N.F.F.E.)
*   Correspondence: missdias@gmail.com

**Abstract:** This paper aims to analyze the strategies that healthcare professionals have adopted during the coronavirus pandemic (COVID-19) in long-term care organizations in Rio de Janeiro city, Brazil, by investigating their competencies—mainly managerial ones. To reach its goals, this paper performs empirical research and theoretical research. For the empirical research, the plans of professionals during COVID-19 pandemic in long-term care organizations are observed, and a questionnaire is applied to analyze observed data integrity. The data are analyzed through the Python and IBM SPSS Statistic programming languages, and descriptive analyses use descriptive statistic proportions, rates, minimum, maximum, mean, median, standard deviation, and coefficient of variation (CV). A non-parametric approach performs repeated measure comparisons using Wilcoxon's test, while the McNemmar test is used to repeat the categorical variables. Statistical significance is assumed at the 5% level. For the theoretical research, a literature review is developed using scientific databases. The results show that for the searched period, the number of deaths and the number of people infected by COVID-19 in these organizations are low when compared to general statistics of Rio de Janeiro city. This paper concludes that these strategical adoptions have brought significant benefits to long-term care organizations, and it might motivate researchers to develop future studies related to long-term care organizations, helping to fill the literature gap on the subject.

**Keywords:** competency; COVID-19; long-term care; management

## 1. Introduction

Brazil has around 28 million people over age 60, which represent 13% of the overall population of the country. By 2025, the elderly population is expected to reach 32 million people, 15% of the overall population of Brazil [1]. Currently, Rio de Janeiro state accounts for 14.5% of the overall elderly population in Brazil [2]. In Rio de Janeiro city, the target of the study, by 2065, the elderly population will represent 36% of the inhabitants, totaling two million people [3]. Long-term care (LTC) organizations are offered to people aged 60 and over of both genders and with different needs and degrees of dependence, who are unable to stay in their homes and their families' homes. In Brazil, there is an increasing demand for organizations that address the needs of elderly people who have weakened or broken family ties; who are experiencing violent situations such as family or organizational neglect; and who are suffering abuse, mistreatment, or loss of capacity for self-care [4,5]. The Brazilian National Health Surveillance Agency (ANVISA) defines LTCs as governmental or non-governmental

residential organizations that are assigned to a household group of people aged 60, or over who have no family support or are in a socially vulnerable situation, offering conditions of freedom and dignity to these individuals [6–8]. The main aim of this paper is to conduct strategical analyses of the ways by which healthcare professionals deal with elderly people in LTCs in Rio de Janeiro city, Brazil during the coronavirus pandemic (COVID-19). Interrupted contact is one of the main coronavirus-related challenges [9]. The World Health Organization (WHO) summarizes COVID-19 surveillance recommendations [10], and it advises that to avoid contagion, in addition to other procedures, people should wash their hands frequently; avoid placing their hands on their eyes, mouth, and nose when they are out; and avoid being in crowded places [11]. Older people are more likely to develop a severe illness and a higher lethality rate [10]. COVID-19 is a family of viruses that cause respiratory infections. The new coronavirus agent was discovered on 31 December 2019, after cases were registered in China [12]. The COVID-19 pandemic creates an opportunity to rethink the rules and focus of society [13]. This paper also investigates competencies that are required for professionals to deal with the virus in the healthcare sector, mainly in managerial activities. Including the Introduction Section, this paper has five sections: Section 2 describes the theoretical framework, Section 3 presents the methodology, Section 4 provides the results and the discussion, and Section 5 concludes and presents limitations and directions for further research.

## 2. The Theoretical Framework

The theoretical section aims to present the theoretical research background. This paper considers, in addition to documents obtained from LTCs, which are based on the Brazilian Health Ministry [12] and WHO [14] orientations, literature information through scientific database searches.

### *2.1. The Strategies to Prevent the COVID-19 Pandemic in Long-Term Care Organizations*

LTCs adopt several strategies to prevent the spread of the COVID-19 pandemic, considering the absolute need to minimize contagion and dissemination of the virus among residents, professionals, and caregivers. The Brazilian National Front for Strengthening Long-Term Institutions developed emergency guidelines for organizations that are entitled the National Front for Strengthening Long Stay Institutions for the Elderly [15]. The idea of these guidelines was to reduce the impact of the COVID-19 pandemic by restricting the delivery of packages to organization headquarters where elderly people are treated in the case of contamination, and to tackle the lack of public policy for LTCs. [15]. In addition to the publication, weekly guidance through internet meetings (lives) were carried out by healthcare surveillance authorities and professionals from Rio de Janeiro city and state to answer organizational questions. These instructions are among those presented in the guidelines and were followed by LTCs.

The first recommendation was to promote isolation by suspending external visits to LTCs. Organizations were asked to stimulate other communication networks such as telephone conversation and social media. To promote social distancing, LTCs were required to establish a schedule for the elderly to leave their rooms for walking around common areas, sunbathing, and eating. The expectation is that residents' time in the common areas will be reduced and that they will avoid crowds, ensuring a minimum distance of one or two meters between them.

LTC areas must have available sinks with liquid soap, paper towels, and a 70% filled alcohol gel pack. All caregivers must be trained in hand washing and in applying 70% gel alcohol several times a day. All the residents must be encouraged to wash their hands and to use alcohol gel every two hours, especially before eating and after using the bathroom. LTCs must guide and supervise proper hand hygiene of residents and of professionals, provide suitable conditions for hand hygiene with water and liquid soap, provide a trash can with a lid that opens without manual contact, and assist elderly people who are unable to clean their hands.

It is fundamental to apply a sodium hypochlorite solution with 30% (bleach) to carpets, cloths, and all entrances of LTCs, especially when employees arrive. Handles, handles of wheelchairs, handrails,

and passageways must be cleaned with a 70% liquid alcohol solution twice a day. The coordination must appoint those responsible for hygiene.

For feeding, LTCs must reinforce food and hydration, with a greater fruit supply. For meal dynamics, organizations must insist that smaller groups of residents to go to the cafeteria at different times and that they maintain a distance of one or two meters from each other. It is advisable to install an information poster throughout LTCs regarding the importance of hygiene and distance measures. Organizations must reduce bureaucracy in application of influenza virus A (H1N1) and pneumococcal vaccines in the institutionalized elderly.

In regard to the delivery of packages to LTCs, it is not permitted to deliver them into the interior of LTCs. All packages must be delivered to reception desk one meter away from the employee who receives them. Before storage, their packaging must be cleaned with a 70% liquid alcoholic solution and external packaging used for delivery must be discarded.

LTC employees are only allowed to enter through the main gate. After arrival in LTCs, employees must wash their hands with soap and water. Then, they must go to a locker-room to take a shower and to change clothes and shoes. Before entering, if it is not possible to change shoes, they must wipe their soles with a cloth dipped in bleach at the LTC entrance door, and they must wear a cap and mask. Household clothes and shoes must remain in the contaminated area. All caregivers and other professionals must wear a surgical mask while they are in LTCs. Laboratorial coats and footwear are of exclusive use at LTCs.

At work, employees must wash their hands or use 70% gel alcohol whenever they touch furniture and before and after they take care of each resident. Employees must wear procedural gloves whenever they are in contact with secretion, urine, or feces. If employees have flu symptoms, they should not go to work. Before entering LTCs, employees must measure their own temperatures.

When employees leave LTCs, they must remove gloves, hats, and masks in the locker-room. They must remove clothes and work shoes, and they must put them in an appropriated place. Then, they must wear their own clothes to go home. Employees must measure their own temperatures, and they must put on their mask to go. Having arrived at home, employees must wash their hands with soap and water, go to the bathroom, shower (wash their hair again), and change their clothes and shoes. Street clothes and shoes must remain in the contaminated area.

## 2.2. Competencies for Healthcare Professionals

The Brazilian healthcare market is changing, and these changes force new measures for professional training [16]. Leslie et al. [17] reinforce this idea, affirming that throughout the last few decades, healthcare medical education suffered extensive modifications. These modifications resulted in increasing demands on the faculty to be creative and to acquire new knowledge, skills, and abilities in a short period [17]. According to Sharma and Zodpey [18] there is a gap between supply and demand for trained healthcare managers to work in organizations like hospitals, pharmaceutical companies, and healthcare insurances.

Cruz [19] affirms that current medical education presents systemic problems such as disconnection among the skills and the needs of patients and the population, poor teamwork, and workforce tendencies to act alone and in continuous competition. Santos Junior et al. [20] suggest that curricular fragmentation and separation between various medical education dimensions have led to a debate on the need for strategies to promote a closer alignment between medicine undergraduate course theory and day-to-day work. Howard et al. [21] add that the pressure of improving healthcare services has never been greater, leading to recent focus on defining and enhancing leadership for healthcare systems and services.

Another issue that needs discussion is competency. Dominguez [22] affirms that it is complex learning that requires integration among knowledge, skills, and attitudes in an integral way in order for them to become habitual behavior. Considering the healthcare sector, Horrocks et al. [23] suggest

that literature shows three competency domain types: core competency, technical/clinical competency, and specialist technical/clinical competency.

Lopes et al. [24] developed a study in Timor-Leste that reveals that managerial competencies are fundamental to improve institutional and individual performances and consequentially to improve healthcare service quality. The study highlights the need of continuing professional training in competencies like communication, leadership, human resource management, financial management, problem solving, professionalism, and knowledge of the healthcare environment [24]. Dowling et al. [25] affirm that continuous learning is a practice that all healthcare professionals should have to keep their clinical knowledge of updated, while Drew and Pandit [26] add that the healthcare learning process includes healthcare process improvement and frontline experience. Creation of a training course to teach professionals in managerial competencies [27,28] is one of the strategies to improve healthcare quality.

Picoli et al. [29] mention that competencies related to managerial activities present less satisfactory results when compared to other general competencies. This may be related to a weakness of developing them in learning scenarios and to the reduced interest of physicians in assuming the function of managers or employers. However, attention must be paid to the need to value these competencies in learning scenarios. Students should understand managerial activity importance during graduation, and they should commit themselves to these activities in healthcare services [29].

Kovacic and Rus [30] examine healthcare managerial leadership competency in Slovenia. According to these authors, leaders give strongest emphasis to interpersonal and informational competencies, while regarding decision-making competency, differences between leaders and other employees are not that significant.

This occurs due to the fact that most managers in the Slovenian healthcare system are medical professionals, who see their leadership roles of healthcare organizations as a step in their medical careers rather than a departure from medical practice and an entry into professional management. Physicians tend to return to medical practice once their leadership mandates run out. As a result, healthcare leadership in Slovenia is highly competent in medical areas, but seriously lacking in managerial competencies [30]. Andersson [31] illustrates that medical leadership implies identity struggles when physicians have manager positions because of the different characteristics of social identities of managers and physicians. Major differences are observed between physicians as autonomous individuals in the system and managers as subordinates to the organizational system.

Meanwhile, Freed [32] outlines the ten core competencies that every hospital manager should practice to continuously revalidate her or his commitment and requalify for the position: to send a condolence card, to donate, to experience all shifts, to visit physicians and nurses where they work, to attend clinical forums, to engage with clinical partners, to experience hospital services, to disclose and genuinely reflect on medical errors and futile care, to cooperate for public benefit, and to remain actively on call.

For Fanelli et al. [33], healthcare managers and professionals have the same idea about competencies to act in the sector, which are the following: quality evaluation based on outcomes; enhancement of professional competencies; programming based on process management; and project cost assessment, informal communication style, and participatory leadership. Harrison et al. [34] describe the five qualities of healthcare leaders: the ability to manage and bring about change, collaborative managers and compassionate leaders, the ability to continuous learn, the ability to balance theoretical and practical management, and leadership skills. According to them, these qualities reflect healthcare service contemporary models and reflect the impact of the globalization of healthcare provision.

Ayeleke et al. [35] and Gibbs and Ashill [36] reveal that leadership is professional training program that can improve individual competency and performance by using flexible and multiple training techniques tailored to organizational contexts. Shirey [37] considers agility as an important leadership competency in healthcare systems, Kumar [38] believes that leadership is essential to healthcare

professionals in a modern healthcare setting, and Drew and Pandit [26] assume that healthcare quality improvement is related to management and leadership.

Riet et al. [39] define three views of medical leadership: a strategic leader, a social leader, and an accepted leader. For all of them it is important to have skills such as collaboration, communication, and integrity. Meanwhile, Haller-Hayon et al. [40] show competency statement samples from the leadership domain, namely, leadership skills and behavior and engaging in culture and environment.

Avolio et al. [41] reveal that leadership development interventions can have a positive effect on attitudes, behaviors, and performance of leaders and their followers. Berghout et al. [42] and Karimi et al. [43] identify two leadership types: Type one includes physicians in formal managerial roles, and they are described as either medical management or medical leadership. Type two contains physicians in informal roles, and they are described as medical leadership. Meanwhile, Scavarda et al. [44] believe that managerial models allow interaction between goods and services and that they bring opportunities to add value to initiatives.

The nursing leaders occupy a crucial position in high-quality work in the community sector [45–47]. They can benefit from daily operation guidance from the organization to attain professional goals [48]. Macphee and Suryaprakash [49] demonstrate how a leadership development program can be a successful strategy for enabling first-line nursing leaders to take an active role in leading staff through managerial initiatives. Charette et al. [50] show nursing curriculum competency-based standardized and validated issues: humanism, healthcare promotion, collaboration, clinical leadership, professionalism, scientific rigor, clinical judgment, and care continuity.

Chen et al. [51] list the five interprofessional practical core principles: core interprofessional concepts, roles/responsibilities, communication skills/tools, conflict management/negotiation, and leadership/membership. For Al-Hussami and Hamad [52], a leadership course may enhance readiness of nursing managers for change. Moreover, healthcare policymakers should focus on applications that increase leadership competency and overall satisfaction of nursing managers to support hospital changes and a learning organization [53].

Deyo et al. [53] believe that the role of nurse manager requires specific skills to translate the vision of the patient. Patient engagement is a set of behaviors by the patients, family members, and healthcare professionals and the implementation of organizational policy engagement into practice and procedures that facilitate inclusion of patients and their family as active members of the healthcare team [53,54].

Kantanen et al. [55] highlight that head nurses need to improve their expertise in general competency areas of communication, influencing skill, and services initiation and innovation. Innovation requires human and financial resources [56]. Head nurses should strengthen their special competency regarding research, political and legal issues, finance, and quality. Nursing directors should increase their expertise in general competency areas of communication and influencing skill, as well as in special competency areas of research, quality, and finance [55]. Flynn et al. [57] conclude that it is essential to increase the focus on nursing roles and skills to contribute to the scientific activities regarding healthcare improvement.

Giacomelli et al. [58] defend the existence of hybrid professionals. For them, managerial performance is a supporting mechanism for organizational change in the public sector through goal clarity and purpose, strategic alignment, motivation, and adaptability. Currie et al. [59] define hybrid middle managers as mediators that are enabled to work through sets of ideas belonging to management and to clinical practice.

Mikkonen et al. [60] develop a study that reveals the importance of preparing healthcare professionals to act in a sustainable healthcare market. Educational professional training of waste disposal healthcare is one of the key and main challenges to solving this matter [61]. In addition, Azevedo et al. [62], Ferrer et al. [63], Magon et al. [64], and Scavarda et al. [65] reinforce the importance of including sustainability in practical activities.

Ojekalu et al. [66] declare that healthcare quality can be improved by supportive visionary leadership; proper planning; education/training; availability of resources; effective management of resources, employees, and processes; and collaboration/cooperation between providers. For Meyer [67], healthcare organizations are invested in improving services, mainly combining operational, improvement, and program managerial expertise. For Dias et al. [68], lean philosophy is one of the strategies that focuses on quality improvement and cost reduction. Lean healthcare is still incipient, but it tends to increase due to changes in healthcare sector mentality. Dau et al. [69] corroborate with the idea, affirming that innovating thoughts allow for the inclusion of smart technologies in healthcare organizations. Considering technology matter, Scavarda et al. [70] state that modalities such as e-learning, e-business, e-health, m-health, and m-learning make digital inclusion possible.

Leadership is one of the main competences required for the crisis period. Leaders are those that can convince the group to follow the objectives with commitment [71]. The crisis environment requires special leadership procedures to deal with the situation [72] and leaders to change their methods to meet new challenges [71]. For Brian et al. [73], agile actions and competency from high leadership are necessary to confront crisis, while O'Neill [74], Bell et al. [75], and Obaid et al. [76] affirm that leaders should be trained to deal with crisis situations.

Emotional intelligence is one of the prerequisites to leadership [71,77]. Other requisites required to leaders, mainly in the crisis period are as follows: to provide strong roles and purpose, to share leadership, to communicate, to ensure employee's access to technology; to prioritize employee's emotional stability, to maintain organizational financial health, and to promote organizational resilience [77].

The COVID-19 crisis encouraged institutions to carry out task forces to face the situations as they presented themselves [78] and others to reflect on how to overcome the crisis [79]. Laxton et al. [80] present five key guidelines that must be observed for policy development to overcome the COVID-19 crisis in LTCs: experts in the sector must take part in sectoral policy development; collaboration between healthcare sectors must become a norm; solutions must contemplate all the sizes of the problem; federal policy leadership must be proactive; and the regulatory process of LTC institutions must be restructured.

Chen [81] concludes that the healthcare sector is changing very quickly, and to develop leadership competencies is vital to follow these changes. In addition, for Sandhu [82], leadership influences thinking and actions, and, consequently, it benefits the practice points as patient care. It also improves the ability to handle changes and chaos. Table 1 summarizes competencies found in the theoretical background.

**Table 1.** Competencies found in the theoretical background.

| Competency | Author |
|---|---|
| Attending clinical forums | [32] |
| Balancing management theory and practice | [34] |
| Collaborative managers and compassionate leaders | [34] |
| Communication | [24,33,39,51,55] |
| Community and customer assessment and engagement | [28] |
| Continuous learning | [25,27,28,34] |
| Cooperating for public benefit | [32] |
| Demonstrating personal qualities | [28] |
| Disclosing and genuinely empathizing about medical errors and futile care | [32] |

**Table 1.** *Cont.*

| Competency | Author |
|---|---|
| Donating | [32] |
| Engaging with clinical partners | [32] |
| Engaging in the culture and environment | [40] |
| Experiencing all shifts | [32] |
| Experiencing hospital services | [32] |
| Financial management | [24] |
| Governance and leadership | [28] |
| Human resource management | [28] |
| Improving services | [67] |
| Influencing skill | [55,56] |
| Innovation | [55,56,66,81,82] |
| Leadership | [21,24,26,30,31,33,34,36–43,49–52,66,81,82] |
| Managing and making change | [34] |
| Managing services | [28,34] |
| Organization | [31,35,36,48] |
| Performance management and accountability | [28] |
| Political analysis and dialogue | [28] |
| Political and legal issues | [55,56] |
| Quality | [45–47,55,68] |
| Remaining actively on call | [32] |
| Research | [55,56] |
| Setting direction | [55] |
| Sending a condolence card | [32] |
| Topic strategic thinking and problem solving | [24,28] |
| Visiting physicians and nurses where they work | [32] |
| Waste disposal (sustainability) | [62–65] |
| Working with others | [55] |

## 3. Materials and Methods

In the Material and Methods Section, this paper presents the path trace to achieve the objectives. This research is shared in two parts: the empirical part that includes observational work and a questionnaire, and the theoretical part that consists of database searching.

### 3.1. The Empirical Research

In the first research stage, the authors of this paper developed the work through participation in internet meetings (lives) and phone interviews to observe strategies adopted by LTC professionals to overcome the impacts of COVID-19. To confirm data integrity, the research was developed by contacting LTCs in Rio de Janeiro city through an e-mail questionnaire. On the internet, the authors of this paper obtained a list from the Open University of the Third Age (UNATI) of Rio de Janeiro State University (UERJ), containing 65 LTCs located in Rio de Janeiro city in 2015 [83]. Based on this list, the authors of this paper made phone calls to obtain e-mails of organizations. The e-mail sending period occurred between 15 May and 25 May 2020. The results were a 41.5% return rate, with 32%

representing responses to the questionnaire and 4.6% representing organizations that reported that they could not be of assistance at that particular time. The remaining 4.6% of organizations asked for further information about research, and they had their inquiries answered by phone but ended up not returning afterwards. One LTC provided "zero" for all answers, and, because of this fact, the data of this organization were not considered. The results of this paper are based on a sample of 20 long-term care organizations, and the data were collected 60 days after the first death by COVID-19 in Brazil [84]. All LTCs agreed to take part in the investigation and in having their data published without revealing the organization's name and the name of the professional who answered the questionnaire. The organizations received a thank-you email.

The statistical evaluation uses the IBM SPSS Statistics for Windows (Version 22.0. IBM Corporation, Armonk, NY, USA), and descriptive analyses used descriptive statistics (proportions, rates, minimum, maximum, mean, median, standard deviation, and coefficient of variation (CV)). Comparison of the repeated measures was performed by a non-parametric approach, using Wilcoxon's test, while the McNemmar test was used to repeat the categorical variables. Statistical significance was assumed at the 5% level. The programming language Python provided the graphical constructions.

*3.2. The Theoretical Research*

In the second stage and aiming to give a theoretical background to the study, this paper includes a survey developed from scientific databases from 2015 to 2020. The search considers the research equations: (("health care") OR (healthcare) OR (health care)) AND (skill) OR (competenc*) AND (manage*) OR (techniq) AND (professio*) AND DOCTYPE (ar OR re) AND PUBYEAR ≥ 2015, and it found 1381 papers. The search totals 258 papers after selection of the following subject areas: health professionals, medicine, nursing, business, management, and accounting. Finally, the authors of this paper conducted a quality check and duplication removal, and the search reached 63 papers that were read, analyzed, and summarized. Figure 1 displays a synthesis of the methodology.

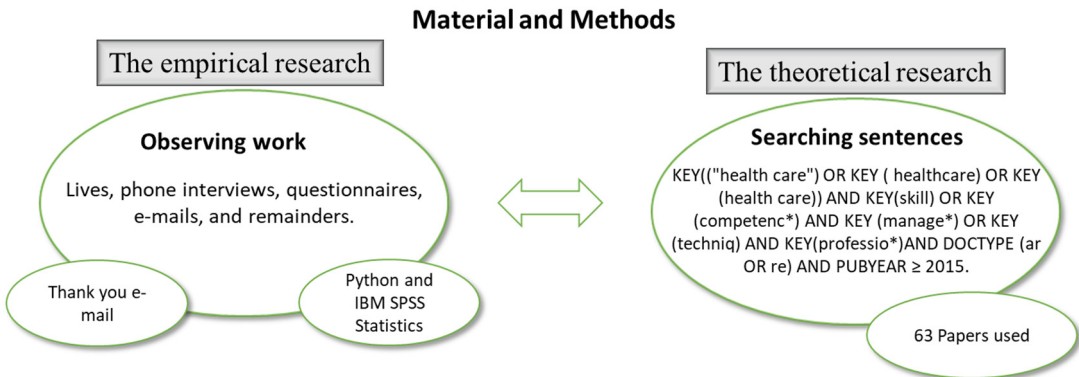

**Figure 1.** Synthesis of the methodology.

## 4. Results

The data analysis of this paper considered the questions 1A, 1B, 2A, 2B, 3A, 3B, and 3C of the questionnaire submitted and answered by LTCs. For information regarding questions 1A, "What is the capacity of your LTC before coronavirus pandemic?" and 1B, "What is the capacity of your LTC after coronavirus pandemic appearance?", see Figure 2.

The 20 organizations are considerably distinct since organization capacity had high variability (CV > 0.4) in the range of from 11 to 80 vacancies, both before and after. The average was 37.10 vacancies before the appearance of the COVID-19 pandemic and 36.15 vacancies afterwards. Organization capacity before and during the COVID-19 pandemic were found to have a *p*-value of 0.042 by using Wilcoxon's test. There is a significant difference between organization capacity before and after the appearance of the COVID-19 pandemic. The 20 organizations had together 742 vacancies before the

pandemic, and capacity decreased to 723 vacancies during the COVID-19 pandemic. The overall capacity reduction rate was 2.56%, which is a statistically significant reduction (see Table 2).

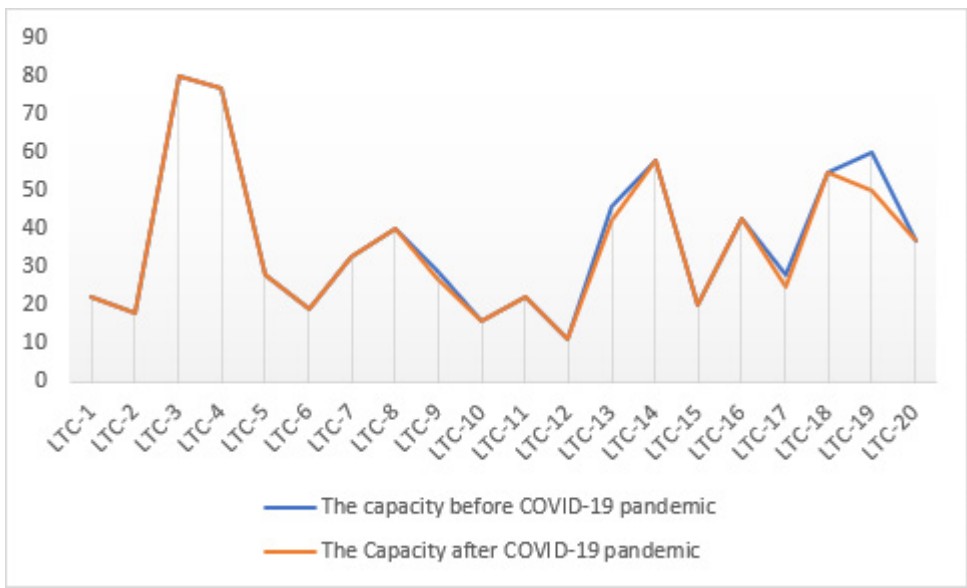

**Figure 2.** The capacity before and after the COVID-19 pandemic.

**Table 2.** The main capacity statistics.

| Variable | Minimum | Maximum | Median | Mean | Standard Deviation | Sum | Coefficient of Variation |
|---|---|---|---|---|---|---|---|
| The capacity before the Coronavirus pandemic | 11 | 80 | 31.00 | 37.10 | 20.07 | 742 | 0.54 |
| The capacity after the appearance of the Coronavirus pandemic | 11 | 80 | 30.50 | 36.15 | 19.63 | 723 | 0.54 |
| The number of residents before the Coronavirus pandemic | 11 | 69 | 27.50 | 31.70 | 18 | 634 | 0.56 |
| The number of residents after the appearance of the Coronavirus pandemic | 11 | 69 | 27.50 | 32.15 | 18 | 643 | 0.56 |
| The occupancy rate before the Coronavirus pandemic | 56.5% | 100.0% | 88.11% | 85.43% | 13.01% | - | 0.15 |
| The occupancy rate after the appearance of the Coronavirus pandemic | 63.2% | 108.0% | 93.30% | 88.22% | 15.06% | - | 0.17 |

See Figure 3 for more information regarding questions 2A, "What is the number of residents at LTC before coronavirus pandemic?" and 2B, "What is the number of residents at LTC after coronavirus pandemic appearance?".

The number of residents in the 20 organizations varied widely in the range from 11 to 69 residents before and after the appearance of the COVID-19 pandemic. When comparing the number of residents before and after the appearance of the coronavirus pandemic, Wilcoxon's test resulted in a *p*-value of 0.573. Therefore, there is no significant difference between the numbers of residents before and after the appearance of the COVID-19 pandemic. In total, organizations had 634 residents before the pandemic and 643 residents after the appearance of the COVID-19 pandemic. The overall increase in the number of residents was 1.42%.

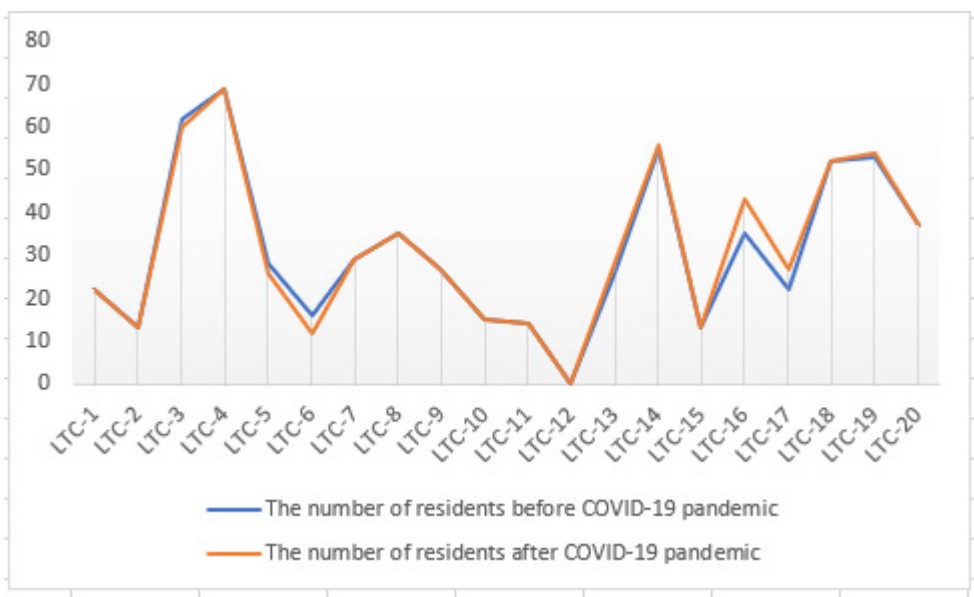

**Figure 3.** The number of the residents before and after the COVID-19 pandemic.

The occupancy rate varied from 56.5% to 100.0% before the pandemic and after the appearance of the COVID-19 pandemic, from 63.2% to 108.0% (one of the organizations had the number of residents as larger than its capacity after the appearance of the coronavirus pandemic). Occupancy rate presents low variability in the sample (CV < 0.20 in both cases, before and after the appearance of the COVID-19 pandemic). Occupation median was 88.1% before the pandemic and 93.3% after the appearance of the COVID-19 pandemic. Occupation mean was 85.43% before the pandemic and 88.22% after the appearance of the coronavirus pandemic. Wilcoxon's test reveals that there is no difference between occupancy rates before and after the pandemic (*p*-value 0.314). Global occupation before the pandemic was 85.44%, and after the appearance of the pandemic, it was 88.94%. See Figures 4–6 for more information regarding questions 3A, "What is the number of infected with coronavirus?"; 3B, "What is the number of suspected cases with coronavirus?"; and 3C, "What is the number of deaths from coronavirus?".

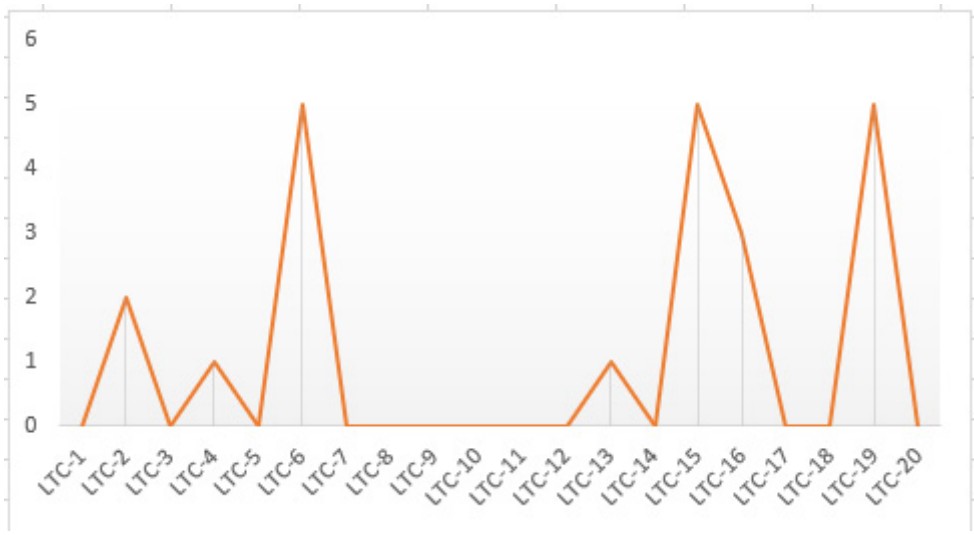

**Figure 4.** The number of people infected with COVID-19.

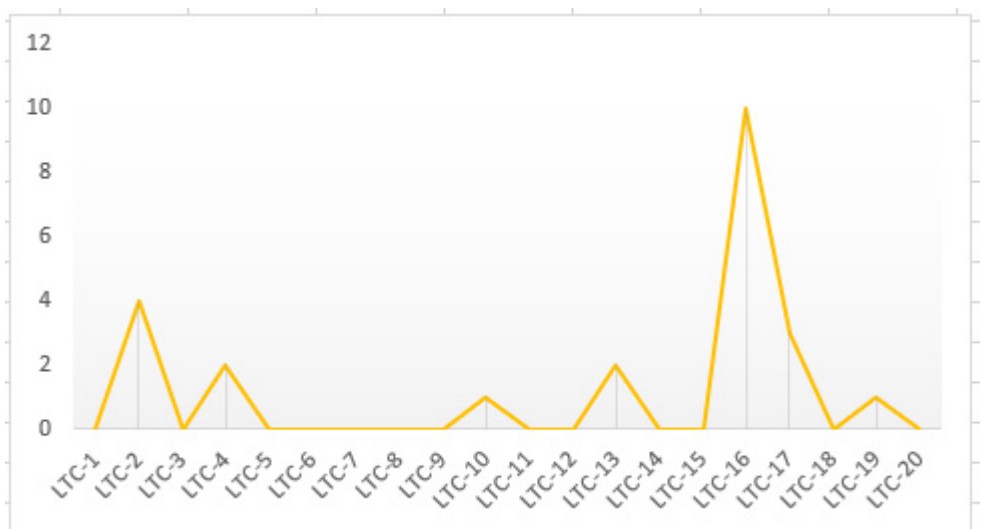

**Figure 5.** The number of suspected cases of COVID-19.

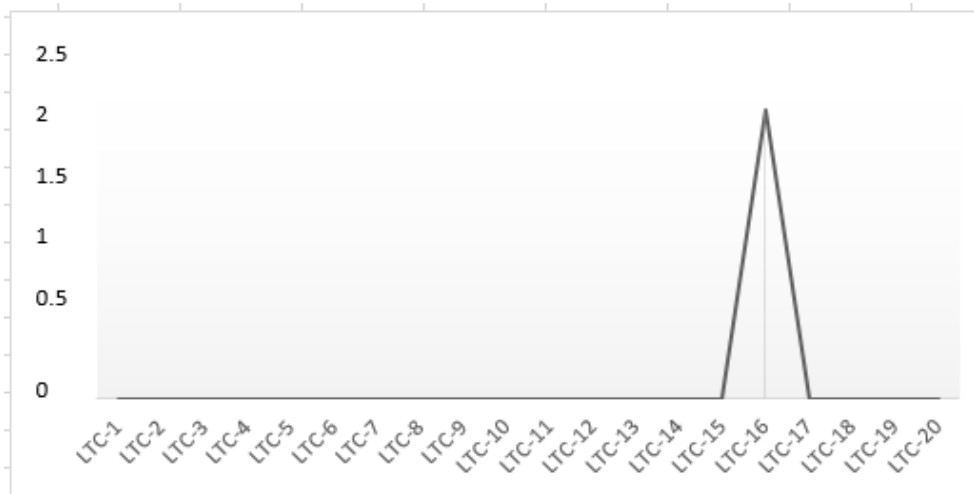

**Figure 6.** The number of COVID-19 deaths.

Among the 20 organizations, 13 (65.0%) had no records regarding the number of people infected with COVID-19. Considering the organizations that had been infected, the number of infected residents ranged from one up to five, and the infection rate ranged from 1.4% to 41.7%. Considering only the organizations that had been infected, there were 22 infected residents out of 232 residents, that is, an infection rate of 9.48%. It was estimated that the global coronavirus infection rate in LTCs was 3.42% (22 out of 643 residents). Among the 20 organizations, 13 (65.0%) had no records regarding suspicion of coronavirus infection. Among those LTCs which had infection suspicion, the number of suspicious ranged from one to ten, and the suspicious rate ranged from 1.9% to 30.8%. There were 23 suspected patients out of 249 patients, resulting in a 9.24% rate of suspicion. It was estimated that the overall suspicion of coronavirus infection rate in LTCs was 3.58% (23 out of 643 residents). Only one organization registered deaths, with two deaths. This organization had 43 residents, three of whom were infected (6.98%), ten suspected (23.26%), and two died (4.65%). Overall, there is a mortality rate of two out of 643. From this sample, it is possible to estimate that the mortality rate due to COVID-19 in long-term care organization, 60 days after the first death by coronavirus in Brazil, was 0.31%; this means among institutionalized elderly people, of every thousand, three died of COVID-19 in Brazil. Infected rate mortality was 9.09% (2 of 22), which means that among institutionalized elderly infected,

of every thousand, 91 died of COVID-19 in Brazil. On 25 May 2020, Rio de Janeiro city registered 22,466 infected people and 2831 deaths due to the COVID-19 pandemic [85,86].

In addition to interviews and observational work, this paper developed a survey of scientific databases to verify in the literature the core managerial competencies required for healthcare professionals. Table 3 lists the result of the survey.

**Table 3.** The core competencies for healthcare professionals.

| Level | Competency | Professional |
|---|---|---|
| Hospital (professional practice) | Leadership | All professionals |
| | Communication | All professionals |
| | Innovation | All professionals |
| | Influencing skill | All professionals |
| | Research | Physicians and nurses |
| | Organization | All professionals |
| | Political and legal issues | Managers (healthcare leaders) |
| | Financial management | Managers (healthcare leaders) |
| | Quality | All professionals |
| | Managing and bringing about change | Managers (healthcare leaders) |
| | Performance management and accountability | Managers (healthcare leaders) |
| | Experiencing all shifts | Managers (healthcare leaders) |
| | Political analysis and dialogue | Managers (healthcare leaders) |
| | Attending clinical forums | Managers (healthcare leaders) |
| | Engaging with clinical partners | Managers (healthcare leaders) |
| | Experiencing hospital services | Managers (healthcare leaders) |
| | Disclosing and genuinely empathizing about medical errors and futile care | Managers (healthcare leaders) |
| | Cooperating for public benefit | Managers (healthcare leaders) |
| | Remaining actively on call | Managers (healthcare leaders) |
| | Balancing management theory and practice | Managers (healthcare leaders) |
| | Sending a condolence card | Managers (healthcare leaders) |
| | Collaborative managers and compassionate leaders | Managers (healthcare leaders) |
| | Continuous learning | All professionals |
| | Topic strategic thinking and problem solving | Managers (healthcare leaders) |
| | Human resource management | Managers (healthcare leaders) |
| | Donating | Managers (healthcare leaders) |
| | Governance and leadership | Managers (healthcare leaders) |
| | Visiting physicians and nurses where they work | Managers (healthcare leaders) |
| | Community and customer assessment and engagement | Managers (healthcare leaders) |
| | Engaging in the culture and environment | Managers (healthcare leaders) |
| | Waste disposal (sustainability) | All professionals |
| | Working with others | Nurses (healthcare leaders) |
| | Demonstrating personal qualities | Nurses (healthcare leaders) |
| | Setting direction | Nurses (healthcare leaders) |
| | Managing services | Nurses (healthcare leaders) |
| | Improving services | Nurses (healthcare leaders) |

The result emphasizes that leadership, innovation, communication, organization, influencing skill, continuous learning, quality, and sustainable competencies are required of all professionals (physicians, nurses, and managers). This study considered the number of times that the competencies appeared in the literature and not the impact caused by each one.

Another point to consider is that the survey adds to healthcare leaders' competency in relation to managerial activities, policies, and activities related to future planning. An expectation is that in addition to the technical part, professional managers in the healthcare sector must have managerial practical knowledge, which requires increasing training. The healthcare sector has several complexities that go well beyond the technological field, thus, understanding the fundamental aspects of managerial work can be a differential between professionals.

Thus, competencies were observed in the group of LTC professionals, especially when they created the front to outline strategies to overcome the COVID-19 pandemic. Leadership, innovation, organization, communication, managerial activities, and trust are competencies identified during the observational work. Figure 7 summarizes the competencies found in the two stages of this paper.

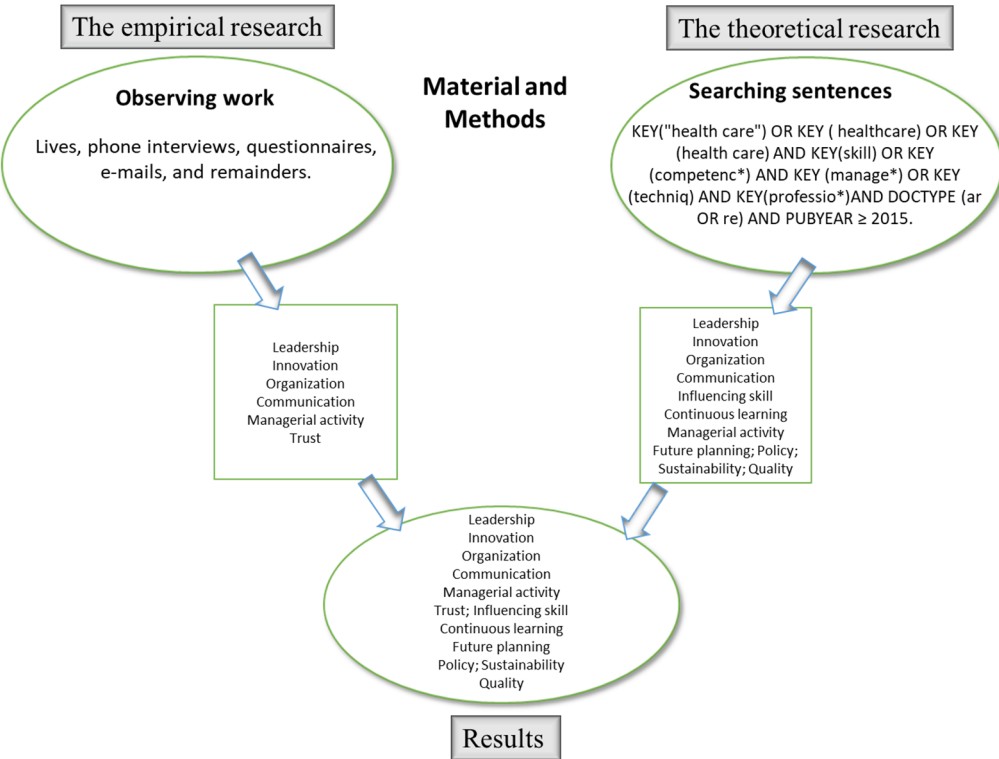

**Figure 7.** Summary of competencies found in the two stages of the study.

## 5. Discussion

This paper shows that the capacity of LTC institutions has decreased by 2.56%, from 742 to 723. This is probably due to the adaptation of LTC's to social distancing. Considering the number of residents in LTCs, an increase of from 634 to 643 residents has been observed, which represents an increase of 1.42%. This shows that the community trusted the strategies adopted by LTCs to protect their residents. Because of these strategies, the number of infected residents was 22, (3.42% when considering the 643 residents), the number of suspicions was 23, (3.58% when considering the 643 residents), and the number of deaths was 2 (0.31% when considering the 643 residents).

The plan developed by professionals to protect elderly people against the COVID-19 pandemic was very successful, and this can be attributed to the competencies and to the commitment of the team with the assignment. Attention should be given to the monitoring carried out by healthcare surveillance and Rio de Janeiro city and state institutions, in particular in regard to the creation of the

Brazilian National Front for Strengthening Long Stay Institutions for the Elderly to combat COVID-19. Elderly people are more susceptible to disease, but care in prevention, especially social isolation, minimized these impacts. Perhaps if others social segments adopted such strategies, the results of the COVID-19 pandemic may be less devastating.

Furthermore, this paper shows that leadership and innovation are the most competencies prominent, especially when during the crisis periods. Professionals must be at the forefront to identify the situations and the actions that need to be taken to minimize the crisis. This highlights the notion of whether training that is given to professionals qualifies them to work in a crisis period.

The COVID-19 pandemic has created a new parameter for the concept of the "normal" in society that is, "new normality." Healthcare professionals, who are directly connected to these changes, must be prepared to improve their competencies and knowledge to support and contribute to these differences. In addition, universities with courses in the healthcare sector must be aware of these changes to better prepare professionals to meet the new market demand. Competency leadership, innovation, organization, communication, managerial activities, trust, influencing skill, continuous learning, sustainability, future planning, policies, and quality raised in this paper should be among the main priorities for organizations that have courses in the healthcare field.

## 6. Conclusions

LTCs have a widely distinct capacity and number of residents, but there is low variability among them in regard to occupation rate. In general analysis, this paper found that long-term organizations had a significant decrease in their capacities after the appearance of the COVID-19 pandemic ($p$-value = 0.042), but the number of residents and occupation rates did not present a significant difference ($p$-values up to 5%). This paper estimates that, with social distancing to combat the COVID-19 pandemic, after 60 days of the first death by COVID-19 in Brazil, in LTCs of Rio de Janeiro city, the infection rate of COVID-19 was 3.42%, the suspicion of COVID-19 infection rate was 3.58%, COVID-19 infection mortality was 9.09%, and the mortality global rate was 0.31%.

The strategical adoptions to improve management brought significant benefits to the analyzed organizations. The plan outlined by LTC professionals saved residents' lives. Healthcare professionals must be aware of competencies that are necessary to develop their activities and to improve their knowledge to face all challenges that are encountered in professional life.

This paper raised competencies that are necessary to work in the healthcare field, mainly in the crisis period. Additionally, this paper has taken long-term care institutions as an example, and it took advantage of the COVID-19 pandemic scenario. The idea is first to intervene in healthcare professionals´ training by ensuring that their training needs meet the needs of the market. In this way, this paper identified leadership, innovation, organization, communication, managerial activity, trust, influencing skill, continuous learning, sustainability, future planning, policy, and quality competencies as the main ones to be considered in professional training. Considering the COVID-19 scenario, this study observed that LTC professionals were very well successful in implementing strategies against the disease.

*Limitations and Directions for Further Research*

This study has limitations regarding the return of the questionnaires. Many LTCs were busy taking care of elderly people and were unable to answer the questionnaire. Another drawback of this analysis is the fact that this research was developed in Rio de Janeiro city, which is located in Rio de Janeiro state in the southeast region of Brazilian. Since Brazil is a continental country, the results of this exploratory research cannot be generalized to all Brazilian cities and states and to other emerging countries. Furthermore, as this is an exploratory study, we did not aim to confirm the hypotheses that were created.

For future research studies, the research results should be generalized by analyzing COVID-19 behavior in LTCs in other Brazilian cities and in other emerging countries, and each healthcare

competency found in the literature review of this paper should be verified by checking the hypotheses that were raised. Another research agenda is to develop research studies that address the theme of Industry 4.0, paying particular attention to social networks. These research studies could survey the impacts of social networks on LTCs before, during, and after the COVID-19 pandemic. Other research studies could address sustainability, that, according to Azevedo et al. [62], is very abstract and requires considerable discussion. These research studies could survey recycling practices in LTCs before, during, and after the COVID-19 pandemic. Sustainable practices are important issues for social policies [87].

**Author Contributions:** Conceptualization, A.D., A.S., A.R., H.S., and N.F.F.E.; formal analysis, A.D., A.S., A.R., H.S., and N.F.F.E.; investigation, A.D., A.S., A.R., H.S., and N.F.F.E.; methodology, A.D., A.S., A.R., H.S., and N.F.F.E.; validation, A.D., A.S., A.R., H.S., and N.F.F.E.; writing—original draft, A.D., A.S., A.R., H.S., and N.F.F.E.; writing—review and editing, A.D., A.S., A.R., H.S., and N.F.F.E. All authors have read and agreed to the published version of the manuscript.

**Funding:** This study was financed in part by the Coordenação de Aperfeiçoamento de Pessoal de Nível Superior—Brasil (CAPES)—Finance Code 001.

**Acknowledgments:** We would like to express our gratitude to all of the LTCs that, despite their busy schedules, collaborated with the study and to all of the professionals who worked hard in an attempt to reduce the impacts of the COVID-19 pandemic.

**Conflicts of Interest:** The authors declare that there is no conflict of interest with the topic addressed.

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
