# Peer review of "Managerial Strategies for Long-Term Care Organization Professionals: COVID-19 Pandemic Impacts"

_sustainability, doi:10.3390/su12229682_

Round 1

Reviewer 1 Report

The topic of the manuscript - strategies adopted by healthcare professionals in Brazil during the Coronavirus pandemic - is of interest to a considerable audience.   While the empirical component generated potentially valuable information through questionnaires, the literature review is too general and is not focused on leadership competencies during times of crisis. Results need to be presented clearly.  And the conclusions need to stem logically from the results.  Extensive editing of English language and style are required.       

Author Response

Dear Reviewer,

We thank you very much for all your comments. Please, see below our answers to your questions and the suggested changes already done.

Kind regards,

The Authors

------------------------------------------------------------------------------------------------------------------------------------------------------------------------

The Reviewer 1 – Comment 1

The topic of the manuscript - strategies adopted by healthcare professionals in Brazil during the Coronavirus pandemic - is of interest to a considerable audience.

Comments of the authors about Reviewer 1 – Comment 1

With this paper, we intend to expand visibility of the themes COVID-19 and long-term care.

------------------------------------------------------------------------------------------------------------------------------------------------------------------------

The Reviewer 1 – Comment 2

While the empirical component generated potentially valuable information through questionnaires, the literature review is too general and is not focused on leadership competencies during times of crisis.

Comments of the authors about Reviewer 1 – Comment 2

We made a new search in scientific databases and found 63 papers with focus on leadership competencies during times of crisis. After checking the quality and relationship with the theme, 12 papers were selected to compose this manuscript.

New wording:

The leadership is one of the main competences required for the crisis period. The leaders are the ones that can convince the group to follow the objectives with commitment [71]. The crisis environment requires the special leadership procedures to deal with the situation [72] and the leaders to change their methods to meet the new challenges [71]. For Brian et al. [73] the agile actions and the competency from the high leadership are necessary to confront the crisis, while O’Neill [74], Bell et al. [75], and Obaid et al. [76] affirm that the leaders should be trained to deal with the crisis situations.

The emotional intelligence is one of the prerequisites to the leadership [71,77]. Others requisites required to the leaders, mainly in the crisis period are: to provide the strong roles and the purpose, to share the leadership, to communicate, to ensure employee’s access to the technology; to prioritize the employee’s emotional stability, to maintain the organizational financial health, and to promote the organizational resilience [77].

The COVID-19 crisis encouraged the institutions to carry out the task forces to face the situations as they presented themselves [78] and the people to reflect how to overcome the crisis [79]. Laxton et al. [80] present the five keys that must be observed for the policy development to overcome the COVID-19 crisis in the LTCs: the experts in the sector must take part in the sectoral policy development; the collaboration between the healthcare sectors must become a norm; the solutions must contemplate all the sizes of the problem; the Federal Policy Leadership must be proactive; and the regulatory process of the LTC institutions must be restructured.

------------------------------------------------------------------------------------------------------------------------------------------------------------------------

The Reviewer 1 – Comment 3

Results need to be presented clearly

Comments of the authors about Reviewer 1 – Comment 3

In the results, we removed the Figures 6, 7, 11 and replaced to a text about them. We believe it is more didactic now.

New wording:

The number of residents in the 20 organizations varied widely in the range from 11 to 69 residents before and after the COVID-19 pandemical appearance. Comparing the number of residents before and after the Coronavirus pandemical appearance, the Wilcoxon’s test resulted in a p-value 0.573. Therefore, there is no significant difference between the numbers of residents before and after the COVID-19 pandemical appearance. In total, the organizations had 634 residents before the pandemic and 643 residents after the COVID-19 pandemical appearance. The overall increasing in the number of residents was 1.42%.

The occupancy rate varied from 56.5% to 100.0% before the pandemic and after the COVID-19 pandemical appearance, from 63.2% to 108.0% (one of the organizations had the number of residents’ lager than its capacity after the Coronavirus pandemical appearance). The occupancy rate presents low variability in the sample (CV<0.20, in both times, before and after the COVID-19 pandemical appearance). The occupation median was 88.1% before the pandemic and 93.3% after the COVID-19 pandemical appearance. The occupation mean was 85.43% before the pandemic and 88.22% after the Coronavirus pandemical appearance. The Wilcoxon’s test reveals that there is no difference between occupancy rates before and after the pandemic (p-value 0.314). The global occupation before the pandemic was 85.44% and after the pandemical appearance was 88.94%.

Between the 20 organizations, 13 (65.0%) had no records of the COVID-19 infected. Considering the organizations that had been infected, the number of infected residents ranged from one up to five and the infection rate ranged from 1.4% up to 41.7%. Considering only the organizations that had been infected, there were 22 infected residents out of 232 residents, that is, an infection rate of 9.48%. It is estimated that the global Coronavirus infection in the LTCs was 3.42% (22 out of 643 residents). Between the 20 organizations, 13 (65.0%) had no Coronavirus infection suspicions records. Those LTCs which had the infection suspicion, the number of the suspicion ranged from one to ten and the suspicions rate ranged from 1.9% up to 30.8%. There were 23 suspicions patients out of 249 patients, resulting in a 9.24% rate of suspicion. It is estimated that the overall Coronavirus infection suspicions rate in the LTCs was 3.58% (23 out of 643 residents). Only one organization registered deaths, with two deaths. This organization had 43 residents, three of whom were infected (6.98%), ten Coronavirus infection suspicion (23.26%), and two deaths (4.65%). Overall, there is a mortality rate of two out of 643. From this sample, it is possible to estimate that the mortality rate due to the COVID-19 in long-term care organization, 60 days after the first death by the Coronavirus in Brazil, was 0.31%; that means, at each thousand institutionalized elderly people, three died of the COVID-19 in Brazil. The mortality of the infected rate is 9.09% (two from 22), that means, at each thousand institutionalized elderly infected, 91 died by the COVID-19 in Brazil. On May 25th, 2020, Rio de Janeiro city registered 22,466 infected people and 2,831 deaths due to the COVID-19 pandemic [85, 86].

------------------------------------------------------------------------------------------------------------------------------------------------------------------------

The Reviewer 1 – Comment 4

The conclusions need to stem logically from the results

Comments of the authors about Reviewer 1 – Comment 4

New wording of our conclusions:

The LTCs have capacity and number of residents widely distinct, but the occupation rate presents low variability between them. In general analysis, this paper found that the long-term organizations had a significant decrease in their capacities after the appearance of the COVID-19 pandemic (p-value 0.042), but the number of residents and the occupation rates did not present significant difference (p-values upper to 5%). This paper estimates that, with the social distance to combat the COVID-19 pandemic, after 60 days of the first death by the COVID-19 in Brazil, in the LTCs of Rio de Janeiro city the COVID-19 infection rate was 3.42%, the COVID-19 infection suspicions rate was 3.58%, the mortality of the COVID-19 infection was 9.09%, and the mortality global rate was 0.31%.

The adoption of the strategies to improve management brings significant benefits to the organizations analyzed. The plan outlined by the LTC professionals saved the residents’ lives. The healthcare professionals must be watchful to the competencies that are necessary to develop their activities and to improve their knowledge to face all the challenges that are proposed by the professional life.

This paper raised the competencies that are necessary to deal in the healthcare sector, mainly in the crisis period. It has taken the long-term care institutions as an example and it took advantage of the COVID-19 pandemical scenario. The idea is first to interfere in the healthcare professionals´ training by identifying that their training needs to meet the needs of the market. In this way, it identified the leadership, innovative, organizational, communication, managerial activity, trust, influencing skill, continuous learning, sustainable, future planning, policy, and quality competencies as the main ones to be considered in a professional training. Considering the COVID-19 scenario, this study observed that the LTC professionals were very well succeed in the strategies against the disease.

This study has limitations regarding the return to the questionnaires. Many LTCs were busy taking care of the elderlies and were unable to answer the questionnaire. Another drawback of this analysis is the fact that this research was developed in Rio de Janeiro city, which is in Rio de Janeiro state, which is in the Brazilian Southeastern region. Since Brazil is a continental country, the results of this exploratory research cannot be generalized to all the Brazilian cities and states and to other emerging countries. Also, as this study is an exploratory one, it did not plan to check the hypotheses that were created.

For future research studies, the prospect is to generalize the research by analyzing the COVID-19 behavior in the LTCs in other Brazilian cities and in other emerging countries and to verify each healthcare competency found in the literature review of this paper by checking the hypotheses that were raised. Another research agenda is to develop research studies that addressed the industry 4.0 theme, considering, especially the social networks. These research studies could survey the impacts of the social networks in the LTCs before, during, and after the COVID-19 pandemic. Other research studies could address the sustainability, that, according to Azevedo et al. [62], is very abstract and needs a deep discussion. These research studies could survey the recycling practices in the LTCs before, during, and after the COVID-19 pandemic. The sustainable practices are important issues for the social policies [87].

------------------------------------------------------------------------------------------------------------------------------------------------------------------------

The Reviewer 1 – Comment 5

Extensive editing of English language and style are required

Comments of the authors about Reviewer 1 – Comment 5

About the English language review, we promoted a review with a native English speaker.

Author Response

REVIEWER 2

Dear Reviewer,

We thank you very much for all your comments. Please, see below our answers to your questions and the suggested changes already done.

Kind regards,

The Authors

------------------------------------------------------------------------------------------------------------------------------------------------------------------------

The Reviewer 2 – Comment 1

This paper makes a valuable contribution by analyzing the strategies adopted by healthcare professionals during COVID-19 in long-term care organizations.

Comments of the authors about Reviewer 2 – Comment 1

With this paper, we intend to expand the discussions on the subject and helping the development of future studies.

------------------------------------------------------------------------------------------------------------------------------------------------------------------------

The Reviewer 2 – Comment 2

In order to reach this goal the authors provided an empirical and theoretical approach to the research.

Comments of the authors about Reviewer 2 – Comment 2

We believe it is important to link the theory with practice.

------------------------------------------------------------------------------------------------------------------------------------------------------------------------

The Reviewer 2 – Comment 3

There are some concerns about the use of English within the paper and the incorrect tense in some areas.

Comments of the authors about Reviewer 2 – Comment 3

About the English language review, we promoted a review with a native English speaker.

------------------------------------------------------------------------------------------------------------------------------------------------------------------------

The Reviewer 2 – Comment 4

I recommend that the authors have the paper proof-read prior to resubmission.

Comments of the authors about Reviewer 2 – Comment 4

We observed and made all the suggested corrections as you can see in the comments.

------------------------------------------------------------------------------------------------------------------------------------------------------------------------

The Reviewer 2 – Comment 5

Line 32: For 2025 the elderly population is expected to reach 32 million

Comments of the authors about Reviewer 2 – Comment 5

The change was successfully made as you can see below.

New wording:

For 2025 the elderly population is expected to reach 32 million people,15% of the Brazilian overall population [1].

------------------------------------------------------------------------------------------------------------------------------------------------------------------------

The Reviewer 2 – Comment 6

Line 39-41: Is this a quote, if so, please add inverted commas. ?

The tendency is a growth of the demand for these organizations in Brazil [3], and the access to the service is guaranteed for elderly people who have weakened or broken family ties, in violent situations like family or organizational neglect, suffering abuse, mistreatment, and loss of the capacity for self-care [4]. The Brazilian National Health Surveillance Agency (ANVISA) defines the LTC as a governmental or non-governmental residential organization that is destined for a home group of people aged 60 or over, with or without family support [5], with condition of freedom and dignity [6], and elderly people in a state of social vulnerability [7].

Comments of the authors about Reviewer 2 – Comment 6

The change was successfully made as you can see below.

New wording:

In Brazil, there is an increasing demand for these organizations that guarantee elderly people who have weakened or broken family ties, who are experiencing the violence situations  like the family or organizational neglect, suffering abuse, mistreatment, or loss of the capacity for self-care [4, 5]. The Brazilian National Health Surveillance Agency (ANVISA) defines the LTC as a governmental or non-governmental residential organization that is assigned to a household group of people aged 60 or over who have or not family support or are in a social vulnerability situation, offering the conditions of freedom and dignity to these individuals [6,7,8].

------------------------------------------------------------------------------------------------------------------------------------------------------------------------

The Reviewer 2 – Comment 7

Line 33: you define Rio de Janeiro as a state, Line 44 as a city. Please clarify and be consistent throughout the manuscript.

Comments of the authors about Reviewer 2 – Comment 7

The change was successfully made as you can see below.

New wording:

Considering Rio de Janeiro city, target of the study, in 2065 the elderly people will represent 36% of the inhabitants, totaling two million people [3].

-----------------------------------------------------------------------------------------------------------------------------------------------------------------------

The Reviewer 2 – Comment 8

Line 153: Consider replacing the word ‘anyway’ with ‘however’

Comments of the authors about Reviewer 2 – Comment 8

The change was successfully made as you can see below.

New wording:

However, attention must be paid to the need to value these competencies in the learning scenarios.

------------------------------------------------------------------------------------------------------------------------------------------------------------------------

The Reviewer 2 – Comment 9

Line 241 … strategies that focuses (remove the word ‘is’)

Comments of the authors about Reviewer 2 – Comment 9

The change was successfully made as you can see below.

New wording:

the strategies that focuses on the quality improvement and the costs reduction.

------------------------------------------------------------------------------------------------------------------------------------------------------------------------

The Reviewer 2 – Comment 10

Line 282/3: and it found 1.381 papers

Comments of the authors about Reviewer 2 – Comment 10

The change was successfully made as you can see below.

New wording:

and it found 1,381 papers.

------------------------------------------------------------------------------------------------------------------------------------------------------------------------

The Reviewer 2 – Comment 11

Line 359: ten suspected

Comments of the authors about Reviewer 2 – Comment 11

The change was successfully made as you can see below.

New wording:

ten suspected (23.26%), and two deaths (4.65%).

Reviewer 3 Report

The authors of the study attempted to present a very important issue in the current situation related to the SARS-CoV-2 pandemic, but the work has serious methodological flaws. The main reference of the research results is the goal and the hypothesis, and this is not the case in the presented work. The discussion requires deep reflection, it is very modest, or rather lack of it. I suggest that the limitations and implications for practice are excluded from the conclusions section and placed as a separate section under discussion. Please specify the conclusions from the research and indicate whether the goal has been achieved and the research hypotheses verified. The abstract does not reflect the structure of the manuscript.

Author Response

REVIEWER 3

Dear Reviewer,

We thank you very much for all your comments. Please, see below our answers to your questions and the suggested changes already done.

Kind regards,

The Authors

------------------------------------------------------------------------------------------------------------------------------------------------------------------------

The Reviewer 3 – Comment 1

The authors of the study attempted to present a very important issue in the current situation related to the SARS-CoV-2 pandemic, but the work has serious methodological flaws. The main reference of the research results is the goal and the hypothesis, and this is not the case in the presented work.

Comments of the authors about Reviewer 3 – Comment 1

The goal of this study is to analyze the strategies adopted by the healthcare professionals during the Coronavirus pandemic (COVID-19) in the long-term care organizations located in the Rio de Janeiro city, Brazil, by investigating their competencies, mainly the managerial ones. The research questions guiding this work are: What are the managerial competences to deal in healthcare sector, mainly in long-term care institutions? Does the appearance of COVID-19 interfere in these competencies? This is an exploratory research. The results found in this study will serve as basis for the development of future studies. In the future studies all the hypotheses raised must be tested.

------------------------------------------------------------------------------------------------------------------------------------------------------------------------

The Reviewer 3 – Comment 2

The discussion requires deep reflection, it is very modest, or rather lack of it.

The plan developed by the professionals to protect the elderly against the COVID-19 pandemic was very well succeed, and it can be attributed to the competencies and to the commitment of the team with the assignment. Highlight can be given to the monitoring carried out by the health surveillance and the Rio de Janeiro state and the city organisms, in particular to the creation of the Brazilian National Front for Strengthening Long Stay Institutions for the Elderly to combat the COVID-19. The elderlies are more susceptible to the disease but care in prevention, specially the social isolation minimized these impacts.

The COVID-19 pandemic has created a new parameter for the normal in the society “New Normality.” The healthcare professionals, who are directly connected to these changes, must be prepared to improve their competencies and knowledge to support and to contribute with these differences. In addition, the universities with courses in the healthcare sector must be aware of these changes to better prepare the professionals to meet the new market demand. The frameworks leadership, innovation, organization, communication, managerial activities, trust, influencing skills, continuous learning, sustainability, future planning, policies, and quality, raised in this paper, should be between the main priorities for the organizations that have courses in the healthcare area.

Comments of the authors about Reviewer 3 – Comment 2

The change was successfully made as you can see below.

New wording:

This paper shows that the capacity of the LTC institutions has decreased 2.56% from 742 to 723. It is probably due to the LTC’s adequation to the social distancing. Considering the number of residents in the LTCs, it has had an increase from 634 to 643 residents, what represents 1.42%. It shows that the community trusted in the strategies adopted by the LTCs to protect their residents. Because of these strategies the number of the infected residents was 22, 3.42% considering the 643 residents, the number of suspicions suspicion was 23, 3.58% considering the 643 residents, and the number of deaths was 2, 0.31% considering the 643 residents.

The plan developed by the professionals to protect the elderly against the COVID-19 pandemic succeed very well and it can be attributed to the competencies and to the commitment of the team with the assignment. Highlight can be given to the monitoring carried out by the healthcare surveillance and the Rio de Janeiro city and state institutions, in particular to the creation of the Brazilian National Front for Strengthening Long Stay Institutions for the Elderly to combat the COVID-19. The elderlies are more susceptible to the disease, but care in prevention, specially the social isolation minimized these impacts. Perhaps if others social segments adopted strategies like these, the COVID-19 pandemical results would be less devastating.

Also, this paper shows that the leadership and the innovation are the competencies most prominent, especially when the crisis periods come. The professionals must be at the forefront to identify the situations and the actions that need to be taken to minimize the crisis. It brings the reflection whether the training that is given to the professionals qualifies them to work in the crisis period.

The COVID-19 pandemic has created a new parameter for the normal in the society “New Normality.” The healthcare professionals, who are directly connected to these changes, must be prepared to improve their competencies and knowledge to support and to contribute with these differences. In addition, the universities with courses in the healthcare sector must be aware of these changes to better prepare the professionals to meet the new market demand. The competencies leadership, innovation, organization, communication, managerial activities, trust, influencing skill, continuous learning, sustainability, future planning, policies, and quality raised in this paper should be among the main priorities for the organizations that have courses in the healthcare area.

------------------------------------------------------------------------------------------------------------------------------------------------------------------------

The Reviewer 3 – Comment 3

I suggest that the limitations and implications for practice are excluded from the conclusions section and placed as a separate section under discussion.

Comments of the authors about Reviewer 3 – Comment 3

The change was successfully made as you can see below.

New wording:

This study has limitations regarding the return to the questionnaires. Many LTCs were busy taking care of the elderlies and were unable to answer the questionnaire. Another drawback of this analysis is the fact that this research was developed in Rio de Janeiro city, which is in Rio de Janeiro state, which is in the Brazilian Southeastern region. Since Brazil is a continental country, the results of this exploratory research cannot be generalized to all the Brazilian cities and states and to other emerging countries. Also, as this study is an exploratory one, it did not plan to check the hypotheses that were created.

------------------------------------------------------------------------------------------------------------------------------------------------------------------------

The Reviewer 3 – Comment 4

Please specify the conclusions from the research and indicate whether the goal has been achieved and the research hypotheses verified.

Comments of the authors about Reviewer 3 – Comment 4

About the hypotheses, please, see our comments to your comment number one.

New wording of our conclusions:

The LTCs have capacity and number of residents widely distinct, but the occupation rate presents low variability between them. In general analysis, this paper found that the long-term organizations had a significant decrease in their capacities after the appearance of the COVID-19 pandemic (p-value 0.042), but the number of residents and the occupation rates did not present significant difference (p-values upper to 5%). This paper estimates that, with the social distance to combat the COVID-19 pandemic, after 60 days of the first death by the COVID-19 in Brazil, in the LTCs of Rio de Janeiro city the COVID-19 infection rate was 3.42%, the COVID-19 infection suspicions rate was 3.58%, the mortality of the COVID-19 infection was 9.09%, and the mortality global rate was 0.31%.

The adoption of the strategies to improve management brings significant benefits to the organizations analyzed. The plan outlined by the LTC professionals saved the residents’ lives. The healthcare professionals must be watchful to the competencies that are necessary to develop their activities and to improve their knowledge to face all the challenges that are proposed by the professional life.

This paper raised the competencies that are necessary to deal in the healthcare sector, mainly in the crisis period. It has taken the long-term care institutions as an example and it took advantage of the COVID-19 pandemical scenario. The idea is first to interfere in the healthcare professionals´ training by identifying that their training needs to meet the needs of the market. In this way, it identified the leadership, innovative, organizational, communication, managerial activity, trust, influencing skill, continuous learning, sustainable, future planning, policy, and quality competencies as the main ones to be considered in a professional training. Considering the COVID-19 scenario, this study observed that the LTC professionals were very well succeed in the strategies against the disease.

This study has limitations regarding the return to the questionnaires. Many LTCs were busy taking care of the elderlies and were unable to answer the questionnaire. Another drawback of this analysis is the fact that this research was developed in Rio de Janeiro city, which is in Rio de Janeiro state, which is in the Brazilian Southeastern region. Since Brazil is a continental country, the results of this exploratory research cannot be generalized to all the Brazilian cities and states and to other emerging countries. Also, as this study is an exploratory one, it did not plan to check the hypotheses that were created.

For future research studies, the prospect is to generalize the research by analyzing the COVID-19 behavior in the LTCs in other Brazilian cities and in other emerging countries and to verify each healthcare competency found in the literature review of this paper by checking the hypotheses that were raised. Another research agenda is to develop research studies that addressed the industry 4.0 theme, considering, especially the social networks. These research studies could survey the impacts of the social networks in the LTCs before, during, and after the COVID-19 pandemic. Other research studies could address the sustainability, that, according to Azevedo et al. [62], is very abstract and needs a deep discussion. These research studies could survey the recycling practices in the LTCs before, during, and after the COVID-19 pandemic. The sustainable practices are important issues for the social policies [87].

------------------------------------------------------------------------------------------------------------------------------------------------------------------------

The Reviewer 3 – Comment 5

The abstract does not reflect the structure of the manuscript.

Abstract: This paper aims to analyze the strategies adopted by the healthcare professionals during the Coronavirus pandemic (COVID-19) in the long-term care organizations located in the Rio de Janeiro city, Brazil, by investigating their competencies, mainly the managerial ones. To reach its goals this paper makes an empirical research and a theoretical research. In the empirical research it observes the plans carried out by the professionals during the COVID-19 pandemic in the long-term care organizations and it applies a questionnaire to analyze the integrity of the observed data. The data are analyzed through the programming languages Python and IBM SPSS Statistics and the descriptive analyses use the descriptive statistics proportions, rates, minimum, maximum, mean, median, standard deviation, and coefficient of variation-CV. In the theoretical research it develops a literature review thought scientific databases. As results, it presents that, in the searched period, the number of deaths and the people infected by the COVID-19 in these organizations were low if compared to the general statistics of the Rio de Janeiro city. This paper concludes that the adoption of these strategies has brought significant benefits to the long-term care organizations. It will support researchers for the development of future studies related to the long-term care organizations, helping to fill the literature gap on the subject.

Comments of the authors about Reviewer 3 – Comment 5

The change was successfully made as you can see below.

New wording:

Abstract: This paper aims to analyze the strategies adopted by the healthcare professionals during the Coronavirus pandemic (COVID-19) in the long-term care organizations located in the Rio de Janeiro city, Brazil, by investigating their competencies, mainly the managerial ones. To reach its goals this paper makes an empirical research and a theoretical research. In the empirical research it observes the plans of the professionals during the COVID-19 pandemic in the long-term care organizations and it applies a questionnaire to analyze the observed datum integrity. The data are analyzed through the Python and IBM SPSS Statistic programming languages and the descriptive analyses use the descriptive statistic proportions, the rates, the minimum, the maximum, the mean, the median, the standard deviation, and the coefficient of variation-CV. A non-parametric approach performs repeated measure comparisons, using the Wilcoxon’s test, while the McNemmar test is used to repeat the categorical variables. The statistical significance assumed at the 5% level. In the theoretical research it develops a literature review thought the scientific databases. As results, it presents that, in the searched period, the number of the deaths and the number of the people infected by the COVID-19 in these organizations are low, if compared to the general statistics of the Rio de Janeiro city. This paper concludes that these strategical adoptions have brought significant benefits to the long-term care organizations. It will stimulate the researchers to develop the future studies related to the long-term care organizations, helping to fill the literature gap on the subject.

Reviewer 4 Report

Abstract

  1. Line 21: Please specify what statistical analyses that you conducted in this study.

Introduction

  1. Line 31: Brazil has around 28 million people aged over 60, what represents 13% of…

→ Brazil has around 28 million people aged over 60, which represents 13% of…

Theoretical framework

  1. 2.2 The competencies for the healthcare professionals: You need to reorganize this section to be more concise and to the point. Currently, you listed different definitions of leadership across a wild range of literature. However, you did not organize and summarize the structure well and it makes the reader get lost in this section and don’t what you want to focus on. Please go through your literature thoroughly, digest all background information and condense and summarize the findings from your literature review.
  2. 3.1 The empirical research:
  • How did you create your rationale? What is your questionnaire? How did you make sure the reliability and validity of the questionnaire? Which theoretical framework did you use to create the questionnaire?
  • What were your independent and dependent variables?
  • What were your research questions?
  • What was the power and effect size of this study? How did you make sure your sample size is large enough to have enough power for statistical analyses? Please provide more information to address these issues.
  • How did you recruit the participants and conduct informed consent process?
  1. 3.2 The theoretical research: Please use the PRISMA guideline to describe your review process.

Discussion

  1. Current information in this section is devoid of content, and you need to look into your study findings carefully, interpret your findings properly, and provide more insights in the discussion by comparing your findings to existing literature.
  2. There were ample limitations of study design that should be mentioned, such as sampling strategy, participant selection, etc. Please provide more information in this section.
  3. What is the uniqueness generated from your study? You have to discuss more information relevant to your research question and provide concrete suggestions to enrich existing knowledge of patient care.

Figure

  1. Figure 2, 3: Please merge these two figures in one by using different lines or colors to present your data.
  1. Figure 4, 5: Please merge these two figures in one by using different lines or colors to present your data.
  1. Figure 6, 7, 11: You can describe the figure information with texts. Please remove these figures.

Author Response

REVIEWER 4

Dear Reviewer,

We thank you very much for all your comments. Please, see below our answers to your questions and the suggested changes already done.

Kind regards,

The Authors

------------------------------------------------------------------------------------------------------------------------------------------------------------------------

The Reviewer 4 – Comment 1

Abstract - Line 21: Please specify what statistical analyses that you conducted in this study.

Abstract: This paper aims to analyze the strategies adopted by the healthcare professionals during the Coronavirus pandemic (COVID-19) in the long-term care organizations located in the Rio de Janeiro city, Brazil, by investigating their competencies, mainly the managerial ones. To reach its goals this paper makes an empirical research and a theoretical research. In the empirical research it observes the plans carried out by the professionals during the COVID-19 pandemic in the long-term care organizations and it applies a questionnaire to analyze the integrity of the observed data. The data are analyzed through the programming languages Python and IBM SPSS Statistics and the descriptive analyses use the descriptive statistics proportions, rates, minimum, maximum, mean, median, standard deviation, and coefficient of variation-CV. In the theoretical research it develops a literature review thought scientific databases. As results, it presents that, in the searched period, the number of deaths and the people infected by the COVID-19 in these organizations were low if compared to the general statistics of the Rio de Janeiro city. This paper concludes that the adoption of these strategies has brought significant benefits to the long-term care organizations. It will support researchers for the development of future studies related to the long-term care organizations, helping to fill the literature gap on the subject.

Comments of the authors about Reviewer 4 – Comment 1

The change was successfully made as you can see below.

Abstract: This paper aims to analyze the strategies adopted by the healthcare professionals during the Coronavirus pandemic (COVID-19) in the long-term care organizations located in the Rio de Janeiro city, Brazil, by investigating their competencies, mainly the managerial ones. To reach its goals this paper makes an empirical research and a theoretical research. In the empirical research it observes the plans of the professionals during the COVID-19 pandemic in the long-term care organizations and it applies a questionnaire to analyze the observed datum integrity. The data are analyzed through the Python and IBM SPSS Statistic programming languages and the descriptive analyses use the descriptive statistic proportions, the rates, the minimum, the maximum, the mean, the median, the standard deviation, and the coefficient of variation-CV. A non-parametric approach performs repeated measure comparisons, using the Wilcoxon’s test, while the McNemmar test is used to repeat the categorical variables. The statistical significance assumed at the 5% level. In the theoretical research it develops a literature review thought the scientific databases. As results, it presents that, in the searched period, the number of the deaths and the number of the people infected by the COVID-19 in these organizations are low, if compared to the general statistics of the Rio de Janeiro city. This paper concludes that these strategical adoptions have brought significant benefits to the long-term care organizations. It will stimulate the researchers to develop the future studies related to the long-term care organizations, helping to fill the literature gap on the subject.

------------------------------------------------------------------------------------------------------------------------------------------------------------------------

The Reviewer 4 – Comment 2

Introduction - Line 31: Brazil has around 28 million people aged over 60, what represents 13% of…

Comments of the authors about Reviewer 4 – Comment 2

The change was successfully made as you can see below.

New wording:

Brazil has around 28 million people aged over 60, which represents 13% of the overall population of the country.

 -----------------------------------------------------------------------------------------------------------------------------------------------------------------------The Reviewer 4 – Comment 3

Theoretical framework - The competencies for the healthcare professionals: You need to reorganize this section to be more concise and to the point. Currently, you listed different definitions of leadership across a wild range of literature. However, you did not organize and summarize the structure well and it makes the reader get lost in this section and don’t what you want to focus on. Please go through your literature thoroughly, digest all background information and condense and summarize the findings from your literature review. The empirical research: How did you create your rationale? What is your questionnaire? How did you make sure the reliability and validity of the questionnaire? Which theoretical framework did you use to create the questionnaire? What were your independent and dependent variables? What were your research questions? What was the power and effect size of this study? How did you make sure your sample size is large enough to have enough power for statistical analyses? Please provide more information to address these issues. How did you recruit the participants and conduct informed consent process?  The theoretical research: Please use the PRISMA guideline to describe your review process.

Comments of the authors about Reviewer 4 – Comment 3

We made a table to summarize our findings.

Table 1. The competencies found in the theoretical background.

Competency

Author

Attend clinical forums

[32]

Balancing management theory and practice

[34]

Collaborative managers and compassionate leaders

[34]

Communication

[24],[33],[39],[51],[55],[56]

Community and customer assessment and engagement

[28]

Continuous learning

[25],[27],[28],[34]

Cooperate for public benefit

[32]

Demonstrating personal qualities

[28]

Disclose and genuinely empathize about medical errors and futile care

[32]

Donate

[32]

Engage clinical partners

[32]

Engaging culture and environment

[40]

Experience all shifts

[32]

Experience hospital services

[32]

Financial management

[24]

Governance and leadership

[28]

Human resource management

[28]

Improving services

[67]

Influencing skill

[55],[56]

Innovation

[55],[56],[66],[81],[82]

Leadership

[21],[24],[26] [30],[31],[33],[34],[36],[37],

[38],[39],[40],[41],[42],[43],[49],[50],[51],[52],[66],[81],[82]

Managing and making change

[34]

Managing services

[28],[34]

Organization

[31],[35],[36],[48]

Performance management and accountability

[28]

Political analysis and dialogue

[28]

Political and legal issues

[55],[56]

Quality

[45],[46],[47],[55],[68]

Remain actively on-call

[32]

Research

[55],[56]

Setting direction

[55]

To send a condolence card

[32]

Topic strategic thinking and problem solving

[24],[28]

Visit physicians and nurses where they work

[32]

Waste disposal (sustainability)

[62],[63],[64],[65]

Working with others

[55]

The goal of this study is to analyze the strategies adopted by the healthcare professionals during the Coronavirus pandemic (COVID-19) in the long-term care organizations located in the Rio de Janeiro city, Brazil, by investigating their competencies, mainly the managerial ones. The research questions guiding this work are: What are the managerial competences to deal in healthcare sector, mainly in long-term care institutions? Does the appearance of COVID-19 interfere in these competencies? We developed a search in the scientific database considering mainly the terms competencies and healthcare. Based on this search we developed the questionnaire that was sent to the LTC institutions, which authorized the publication of the results, as you can see below.

The questionaire

Dear LTC

I' am Ana Claudia Dias doctoral degree student at the Federal Center for Technological Education Celso Suckow da Fonseca - CEFET/RJ. I research the Coronavirus pandemic (COVID-19) associated with elderly people living in the long-term care institutions and I invite you to participate in the study by answering the questions below. This study aims to help better understand what is happening in the LTCs and your response in the shortest possible time will help to share the data. If you need, you can send email (missdias@gmail.com) or call (21-999525855) to me. Thank you!

I authorize the results of this study to be presented and published in events and magazines, knowing that my name and my institution will be kept strictly confidential. ( ) Yes No ( )

1a- What is the capacity of your LTC before the COVID-19 pandemic?

1b -What is the capacity of your LTC after the COVID-19 pandemical appearance?

2a-What is the number of residents at the LTC before the COVID-19 pandemic?

2b-What is the number of residents at the LTC after the COVID-19 pandemical appearance?

3a-What is the number of infected with the COVID-19?

3b-What is the number of suspected cases with the COVID-19?

3c-What is the number of deaths from the COVID-19?

4a-How many residents used the social network before the COVID-19 pandemic?

4b-How many residents use the social network after the COVID-19 pandemical appearance?

5a-How many residents participated in recreational and socializing activities before the COVID-19 pandemic?

5b- How many residents participate in recreational and socializing activity after the COVID-19 pandemical appearance?

6a-Did your LTC practice recycling before the COVID-19 pandemic? ( ) Yes No ( )

6b-Does your LTC practice recycling after the COVID-19 pandemical appearance?

() Yes No ()

The institutions that participated in the investigation received a thank you email.The thank you email

Dear LTC

Thank you very much for your contribution. Your participation was extremely important for our research. You will receive the results of the study soon.

Best regards,

Ana Claudia Dias

This is an exploratory research. The results found in this study will serve as basis for the development of future studies. In the future studies all the hypotheses raised must be tested.

-----------------------------------------------------------------------------------------------------------------------------------------------------------------------

The Reviewer 4 – Comment 4

Discussion - Current information in this section is devoid of content, and you need to look into your study findings carefully, interpret your findings properly, and provide more insights in the discussion by comparing your findings to existing literature.

The plan developed by the professionals to protect the elderly against the COVID-19 pandemic was very well succeed, and it can be attributed to the competencies and to the commitment of the team with the assignment. Highlight can be given to the monitoring carried out by the health surveillance and the Rio de Janeiro state and the city organisms, in particular to the creation of the Brazilian National Front for Strengthening Long Stay Institutions for the Elderly to combat the COVID-19. The elderlies are more susceptible to the disease but care in prevention, specially the social isolation minimized these impacts.

The COVID-19 pandemic has created a new parameter for the normal in the society “New Normality.” The healthcare professionals, who are directly connected to these changes, must be prepared to improve their competencies and knowledge to support and to contribute with these differences. In addition, the universities with courses in the healthcare sector must be aware of these changes to better prepare the professionals to meet the new market demand. The frameworks leadership, innovation, organization, communication, managerial activities, trust, influencing skills, continuous learning, sustainability, future planning, policies, and quality, raised in this paper, should be between the main priorities for the organizations that have courses in the healthcare area.

Comments of the authors about Reviewer 4 – Comment 4

The change was successfully made as you can see below.

New wording:

This paper shows that the capacity of the LTC institutions has decreased 2.56% from 742 to 723. It is probably due to the LTC’s adequation to the social distancing. Considering the number of residents in the LTCs, it has had an increase from 634 to 643 residents, what represents 1.42%. It shows that the community trusted in the strategies adopted by the LTCs to protect their residents. Because of these strategies the number of the infected residents was 22, 3.42% considering the 643 residents, the number of suspicions suspicion was 23, 3.58% considering the 643 residents, and the number of deaths was 2, 0.31% considering the 643 residents.

The plan developed by the professionals to protect the elderly against the COVID-19 pandemic succeed very well and it can be attributed to the competencies and to the commitment of the team with the assignment. Highlight can be given to the monitoring carried out by the healthcare surveillance and the Rio de Janeiro city and state institutions, in particular to the creation of the Brazilian National Front for Strengthening Long Stay Institutions for the Elderly to combat the COVID-19. The elderlies are more susceptible to the disease, but care in prevention, specially the social isolation minimized these impacts. Perhaps if others social segments adopted strategies like these, the COVID-19 pandemical results would be less devastating.

Also, this paper shows that the leadership and the innovation are the competencies most prominent, especially when the crisis periods come. The professionals must be at the forefront to identify the situations and the actions that need to be taken to minimize the crisis. It brings the reflection whether the training that is given to the professionals qualifies them to work in the crisis period.

The COVID-19 pandemic has created a new parameter for the normal in the society “New Normality.” The healthcare professionals, who are directly connected to these changes, must be prepared to improve their competencies and knowledge to support and to contribute with these differences. In addition, the universities with courses in the healthcare sector must be aware of these changes to better prepare the professionals to meet the new market demand. The competencies leadership, innovation, organization, communication, managerial activities, trust, influencing skill, continuous learning, sustainability, future planning, policies, and quality raised in this paper should be among the main priorities for the organizations that have courses in the healthcare area.

------------------------------------------------------------------------------------------------------------------------------------------------------------------------

The Reviewer 4 – Comment 5

There were ample limitations of study design that should be mentioned, such as sampling strategy, participant selection, etc. Please provide more information in this section.

Comments of the authors about Reviewer 4 – Comment 5

We included a paragraph in the conclusions to emphasize the study limitations, as you can see below.

New wording:

The LTCs have capacity and number of residents widely distinct, but the occupation rate presents low variability between them. In general analysis, this paper found that the long-term organizations had a significant decrease in their capacities after the appearance of the COVID-19 pandemic (p-value 0.042), but the number of residents and the occupation rates did not present significant difference (p-values upper to 5%). This paper estimates that, with the social distance to combat the COVID-19 pandemic, after 60 days of the first death by the COVID-19 in Brazil, in the LTCs of Rio de Janeiro city the COVID-19 infection rate was 3.42%, the COVID-19 infection suspicions rate was 3.58%, the mortality of the COVID-19 infection was 9.09%, and the mortality global rate was 0.31%.

The adoption of the strategies to improve management brings significant benefits to the organizations analyzed. The plan outlined by the LTC professionals saved the residents’ lives. The healthcare professionals must be watchful to the competencies that are necessary to develop their activities and to improve their knowledge to face all the challenges that are proposed by the professional life.

This paper raised the competencies that are necessary to deal in the healthcare sector, mainly in the crisis period. It has taken the long-term care institutions as an example and it took advantage of the COVID-19 pandemical scenario. The idea is first to interfere in the healthcare professionals´ training by identifying that their training needs to meet the needs of the market. In this way, it identified the leadership, innovative, organizational, communication, managerial activity, trust, influencing skill, continuous learning, sustainable, future planning, policy, and quality competencies as the main ones to be considered in a professional training. Considering the COVID-19 scenario, this study observed that the LTC professionals were very well succeed in the strategies against the disease.

This study has limitations regarding the return to the questionnaires. Many LTCs were busy taking care of the elderlies and were unable to answer the questionnaire. Another drawback of this analysis is the fact that this research was developed in Rio de Janeiro city, which is in Rio de Janeiro state, which is in the Brazilian Southeastern region. Since Brazil is a continental country, the results of this exploratory research cannot be generalized to all the Brazilian cities and states and to other emerging countries. Also, as this study is an exploratory one, it did not plan to check the hypotheses that were created.

For future research studies, the prospect is to generalize the research by analyzing the COVID-19 behavior in the LTCs in other Brazilian cities and in other emerging countries and to verify each healthcare competency found in the literature review of this paper by checking the hypotheses that were raised. Another research agenda is to develop research studies that addressed the industry 4.0 theme, considering, especially the social networks. These research studies could survey the impacts of the social networks in the LTCs before, during, and after the COVID-19 pandemic. Other research studies could address the sustainability, that, according to Azevedo et al. [62], is very abstract and needs a deep discussion. These research studies could survey the recycling practices in the LTCs before, during, and after the COVID-19 pandemic. The sustainable practices are important issues for the social policies [87].

------------------------------------------------------------------------------------------------------------------------------------------------------------------------

The Reviewer 4 – Comment 6

What is the uniqueness generated from your study? You have to discuss more information relevant to your research question and provide concrete suggestions to enrich existing knowledge of patient care.

Comments of the authors about Reviewer 4 – Comment 6

This paragraph was also included in our conclusion.

This paper raised the competencies that are necessary to deal in the healthcare sector, mainly in the crisis period. It has taken the long-term care institutions as an example and it took advantage of the COVID-19 pandemical scenario. The idea is first to interfere in the healthcare professionals´ training by identifying that their training needs to meet the needs of the market. In this way, it identified the leadership, innovative, organizational, communication, managerial activity, trust, influencing skill, continuous learning, sustainable, future planning, policy, and quality competencies as the main ones to be considered in a professional training. Considering the COVID-19 scenario, this study observed that the LTC professionals were very well succeed in the strategies against the disease.------------------------------------------------------------------------------------------------------------------------------------------------------------------------

The Reviewer 4 – Comment 7

Figure 2, 3: Please merge these two figures in one by using different lines or colors to present your data.

Comments of the authors about Reviewer 4 – Comment 7

The change was successfully made as you can see below.

------------------------------------------------------------------------------------------------------------------------------------------------------------------------

The Reviewer 4 – Comment 8

Figure 4, 5: Please merge these two figures in one by using different lines or colors to present your data.

Comments of the authors about Reviewer 4 – Comment 8

The change was successfully made as you can see below.

------------------------------------------------------------------------------------------------------------------------------------------------------------------------

The Reviewer 4 – Comment 9

Figure 6, 7, 11: You can describe the figure information with texts. Please remove these figures.

Comments of the authors about Reviewer 4 – Comment 9

The figures were removed and replaced by a text about them.

The number of residents in the 20 organizations varied widely in the range from 11 to 69 residents before and after the COVID-19 pandemical appearance. Comparing the number of residents before and after the Coronavirus pandemical appearance, the Wilcoxon’s test resulted in a p-value 0.573. Therefore, there is no significant difference between the numbers of residents before and after the COVID-19 pandemical appearance. In total, the organizations had 634 residents before the pandemic and 643 residents after the COVID-19 pandemical appearance. The overall increasing in the number of residents was 1.42%.

The occupancy rate varied from 56.5% to 100.0% before the pandemic and after the COVID-19 pandemical appearance, from 63.2% to 108.0% (one of the organizations had the number of residents’ lager than its capacity after the Coronavirus pandemical appearance). The occupancy rate presents low variability in the sample (CV<0.20, in both times, before and after the COVID-19 pandemical appearance). The occupation median was 88.1% before the pandemic and 93.3% after the COVID-19 pandemical appearance. The occupation mean was 85.43% before the pandemic and 88.22% after the Coronavirus pandemical appearance. The Wilcoxon’s test reveals that there is no difference between occupancy rates before and after the pandemic (p-value 0.314). The global occupation before the pandemic was 85.44% and after the pandemical appearance was 88.94%.

Between the 20 organizations, 13 (65.0%) had no records of the COVID-19 infected. Considering the organizations that had been infected, the number of infected residents ranged from one up to five and the infection rate ranged from 1.4% up to 41.7%. Considering only the organizations that had been infected, there were 22 infected residents out of 232 residents, that is, an infection rate of 9.48%. It is estimated that the global Coronavirus infection in the LTCs was 3.42% (22 out of 643 residents). Between the 20 organizations, 13 (65.0%) had no Coronavirus infection suspicions records. Those LTCs which had the infection suspicion, the number of the suspicion ranged from one to ten and the suspicions rate ranged from 1.9% up to 30.8%. There were 23 suspicions patients out of 249 patients, resulting in a 9.24% rate of suspicion. It is estimated that the overall Coronavirus infection suspicions rate in the LTCs was 3.58% (23 out of 643 residents). Only one organization registered deaths, with two deaths. This organization had 43 residents, three of whom were infected (6.98%), ten Coronavirus infection suspected (23.26%), and two deaths (4.65%). Overall, there is a mortality rate of two out of 643. From this sample, it is possible to estimate that the mortality rate due to the COVID-19 in long-term care organization, 60 days after the first death by the Coronavirus in Brazil, was 0.31%; that means, at each thousand institutionalized elderly people, three died of the COVID-19 in Brazil. The mortality of the infected rate is 9.09% (two from 22), that means, at each thousand institutionalized elderly infected, 91 died by the COVID-19 in Brazil. On May 25th, 2020, Rio de Janeiro city registered 22,466 infected people and 2,831 deaths due to the COVID-19 pandemic [85, 86].

Round 2

Reviewer 1 Report

Dear Authors, Thank you for taking the time and effort to revise the manuscript on "The Managerial Strategies for the Long-Term Care Organizational Professionals: The COVID-19 Pandemical Impacts".  While the literature search is improved, major issues remain with writing, results, and conclusions, and so in my view the manuscript occupancy rate is used as a direct proxy of success in managing the epidemic.  Quantitative presentation of results were difficult to follow. Nevertheless, I hope that this feedback is helpful and wish you success in finding another outlet.          

Author Response

Dear Reviewer,

We are grateful for your comments. They helped us very much to improve our paper. Our comments to your comments are in blue.

We hope our paper is now suitable for publication.

Thank you once again for the opportunity.

Kind regards,

The Authors

Reviewer 3 Report

I suggest that the limitations and implications for practice are excluded from the conclusions section and placed as a separate section under discussion.

Author Response

Reviewer 3

Dear Reviewer,

We are grateful for your comments. They helped us very much to improve our paper. Our comments to your comments are in blue.

Thank you once again for the opportunity.

Kind regards,

The Authors

The Reviewer 3 – Comment 1

I suggest that the limitations and implications for practice are excluded from the conclusions section and placed as a separate section under discussion.

Comments of the authors about Reviewer 3 – Comment 1

  1. Conclusions

The LTCs have the capacity and the number of residents widely distinct, but the occupation rate presents low variability between them. In general analysis, this paper found that the long-term organizations had a significant decrease in their capacities after the COVID-19 pandemical appearance (p-value 0.042), but the number of residents and the occupation rates did not present significant difference (p-values upper to 5%). This paper estimates that, with the social distance to combat the COVID-19 pandemic, after 60 days of the first death by the COVID-19 in Brazil, in the LTCs of Rio de Janeiro city the COVID-19 infection rate was 3.42%, the COVID-19 infection suspicious rate was 3.58%, the mortality of the COVID-19 infection was 9.09%, and the mortality global rate was 0.31%.

The strategical adoptions to improve the management brought significant benefits to the organizations analyzed. The plan outlined by the LTC professionals saved the residents’ lives. The healthcare professionals must be watchful to the competencies that are necessary to develop their activities and to improve their knowledge to face all the challenges that are proposed by the professional life.

This paper raised the competencies that are necessary to deal in the healthcare field, mainly in the crisis period. It has taken the long-term care institutions as an example and it took advantage of the COVID-19 pandemical scenario. The idea is first to interfere in the healthcare professionals´ training by identifying that their training needs to meet the needs of the market. In this way, it identified the leadership, innovative, organizational, communication, managerial activity, trust, influencing skill, continuous learning, sustainable, future planning, policy, and quality competencies as the main ones to be considered in the professional training. Considering the COVID-19 scenario, this study observed that the LTC professionals were very well succeed in the strategies against the disease.

6.1. - The limitations and the directions for the further research

This study has limitations regarding the return to the questionnaires. Many LTCs were busy taking care of the elderly people and were unable to answer the questionnaire. Another drawback of this analysis is the fact that this research was developed in the Rio de Janeiro city, which is in the Rio de Janeiro state, which is in the Brazilian Southeastern region. Since Brazil is a continental country, the results of this exploratory research cannot be generalized to all the Brazilian cities and states and to other emerging countries. Also, as this study is an exploratory one, it did not plan to check the hypotheses that were created.

For the future research studies, the prospect is to generalize the research by analyzing the COVID-19 behavior in the LTCs in other Brazilian cities and in other emerging countries and to verify each healthcare competency found in the literature review of this paper by checking the hypotheses that were raised. Another research agenda is to develop the research studies that addressed the industry 4.0 theme, considering, especially the social networks. These research studies could survey the impacts of the social networks in the LTCs before, during, and after the COVID-19 pandemic. Other research studies could address the sustainability, that, according to Azevedo et al. [62], is very abstract and needs a deep discussion. These research studies could survey the recycling practices in the LTCs before, during, and after the COVID-19 pandemic. The sustainable practices are important issues for the social policies [87].

Reviewer 4 Report

The authors have fully addressed the reviewer’s comments, and this version improves a lot in contexts and its readability. I am looking forwards to reading this manuscript published in Sustainability.

Author Response

Reviewer 4

The Reviewer 4 – Comment 1

The authors have fully addressed the reviewer’s comments, and this version improves a lot in contexts and its readability. I am looking forwards to reading this manuscript published in Sustainability.

Dear Reviewer,

We are grateful for your comments. After your suggestions were completed, we noticed that our study got even better!

Thank you once again for the opportunity.

Kind regards,

The Authors

Round 3

Reviewer 1 Report

The manuscript is improved substantially.   A remaining issue is that the text in its entirety must be edited for use of the English language (e.g. change "pandemical(sic)" to pandemic).